# Bisguanidinium dinuclear oxodiperoxomolybdosulfate ion pair-catalyzed enantioselective sulfoxidation

Lili Zong[1], Chao Wang[1], Adhitya Mangala Putra Moeljadi[1], Xinyi Ye[1], Rakesh Ganguly[1], Yongxin Li[1], Hajime Hirao[1] & Choon-Hong Tan[1]

Catalytic use of peroxomolybdate for asymmetric transformations has attracted increasing attention due to its catalytic properties and application in catalysis. Herein, we report chiral bisguanidinium dinuclear oxodiperoxomolybdosulfate [**BG**]$^{2+}$[$(\mu\text{-}SO_4)Mo_2O_2(\mu\text{-}O_2)_2(O_2)_2$]$^{2-}$ ion pair, as a catalyst for enantioselective sulfoxidation using aqueous $H_2O_2$ as the terminal oxidant. The ion pair catalyst is isolatable, stable and useful for the oxidation of a range of dialkyl sulfides. The practical utility was illustrated using a gram-scale synthesis of armodafinil, a commercial drug, with the catalyst generated *in situ* from 0.25 mol% of bisguanidinium and 2.5 mol% of $Na_2MoO_4 \cdot 2H_2O$. Structural characterization of this ion pair catalyst has been successfully achieved using single-crystal X-ray crystallography.

[1] Division of Chemistry and Biological Chemistry, School of Physical and Mathematical Sciences, Nanyang Technological University, 21 Nanyang Link, Singapore 637371, Singapore. Correspondence and requests for materials should be addressed to H.H. (email: hirao@ntu.edu.sg) or to C.-H.T. (email: choonhong@ntu.edu.sg).

Metalloenzymes containing molybdenum, responsible for nitrogen and sulfur metabolism[1–4], continue to fuel interest in the exploration of novel molybdenum complexes with catalytic activities. Various neutral coordination complexes of Mo(VI) containing chiral organic ligands have been synthesized and studied extensively for enantioselective reactions[5–11]. In particular, one interesting example of highly enantioselective sulfoxidation of alkyl aryl sulfides was demonstrated using a complex derived from MoO₂(acac)₂ and chiral bis-hydroxamic acids[12]. In contrast to their neutral counterparts, there have been no successful attempts to utilize peroxomolybdate[13] for asymmetric reactions, even though many of these species have been comprehensively characterized.

It is well known that peroxomolybdates are formed on the treatment of molybdate salts with aqueous H₂O₂ oxidant[14,15]. Monomeric, oligomeric and polymeric peroxomolybdate species could be generated under similar conditions (Fig. 1)[14]. The addition of different organic ligands can further increase the structural and functional diversity of peroxomolybdate complexes[16–18]. Other ligands such as silanol[19], phosphate[20], arsenate[21] and sulfate[22–25] have been used to bridge molybdates to construct dinuclear or trinuclear peroxomolybdate complexes. Peroxomolybdates species have been shown to be excellent catalysts for the oxidation of numerous substrates, including alkenes, alcohols[26] and sulfides. The complexity of peroxomolybdates is thus recognized to be a challenging obstacle for elaborating them into highly enantioselective catalysts.

We have recently developed pentanidium[27–30] and dicationic bisguanidinium (**BG**) as efficient phase-transfer[31,32] and ion pair catalysts[33–37]. We have utilized bisguanidinium permanganate ion pair catalyst for the enantioselective oxidation of alkenes[38]. The precise stereocontrol exhibited by bisguanidinium encouraged us to explore other anionic metallic species for asymmetric transformations[39]. Herein, we describe our serendipitous discovery of chiral bisguanidinium dinuclear oxodiperoxomolybdosulfate [**BG**]²⁺[(μ-SO₄)Mo₂O₂(μ-O₂)₂(O₂)₂]²⁻ ion pair catalyst (Fig. 2). This ion pair catalyst is stable and isolatable or it can be generated *in situ*. In a continuation of our current efforts towards developing practical approaches to enantiopure sulfoxides[29,40,41], we report a simple and scalable methodology for enantioselective sulfoxidation using this ion pair catalyst[42–45].

## Results

**Catalytic application of molybdate in sulfoxidation.** At the onset of this work, we realized that we were unable to approach enantiopure 2-sulfinyl esters through enantioselective alkylation of sulfenate anion, as the reaction was incompatible with α-halogenated carboxylates (Supplementary Fig. 1)[29]. We were attracted to the low cost and easy accessibility of molybdate salts and thus we attempted to investigate the direct sulfoxidation of 2-sulfanyl acetate, by utilizing a catalytic amount of molybdate salts and aqueous H₂O₂ as terminal oxidant. Methyl 2-(benzhydrylsulfanyl)acetate **2a** was chosen as the model

substrate (Table 1), since 2-sulfinyl acetate **3a** could be easily transformed to armodafinil, a commercial drug used for the treatment of narcolepsy and shift work sleep disorder[46–48].

When the reaction was performed in the presence of 1 mol% of (*S,S*)-**1a**, 5 mol% of (NH₄)₆Mo₇O₂₄·4H₂O and 1.05 equiv. 35% aqueous H₂O₂, poor yield and no enantioselectivity were observed (Table 1, entry 1). With the addition of acetic acid[49], the enantioselectivity was slightly improved, albeit with low yield (entries 2 and 3). Using trifluoroacetic acid as additive, a marked enhancement of the reactivity was achieved, but with negligible enantioselectivity (entry 4). With the addition of sodium or potassium hydrogen sulfate, we observed significant improvement of yield, as well as enantioselectivity (entries 5 and 6)[50,51]. Switching to other additives, such as dihydrogen phosphate or hydrogen phosphate led to poor results (entries 7 and 8). Further investigation of reaction parameters (entries 9–14), such as the source of molybdate, solvent and stoichiometry of KHSO₄, led to the enhancement of reactivity, as well as enantioselectivity (entry 14, 93% *ee*; Supplementary Table 1). The optimal condition was established by lowering the

**a**

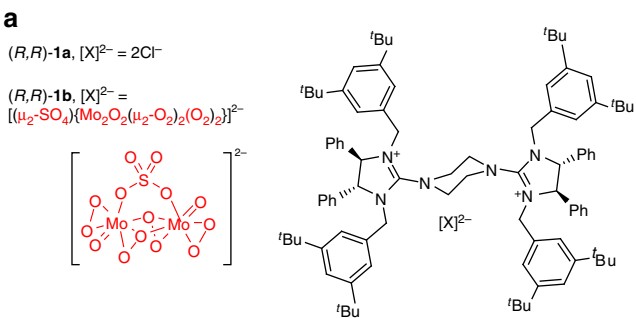

**b**

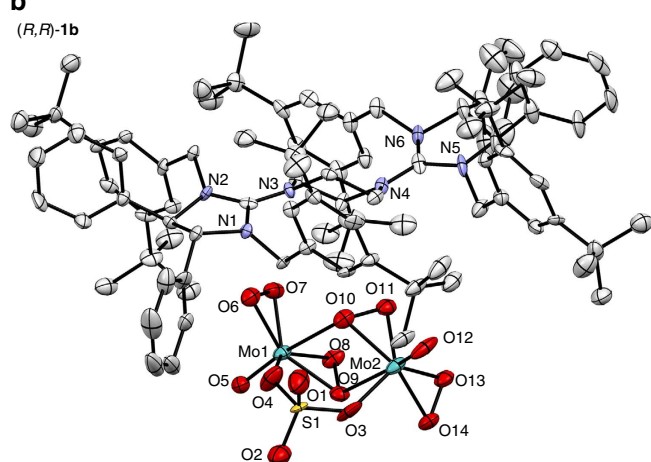

**Figure 2 | Bisguanidinium ion pairs (*R,R*)-1a and (*R,R*)-1b.** (**a**) Structure of bisguanidinium salts. (**b**) X-ray crystallographic structure of [**BG**]²⁺ [(μ-SO₄)Mo₂O₂(μ-O₂)₂(O₂)₂]²⁻ (*R,R*)-**1b** (ellipsoids at 50% probability).

**Figure 1 | Examples of peroxomolybdate complexes.** Monomeric, dimeric, trimeric and tetrameric peroxomolybdates generated by treating molybdate with H₂O₂.

**Table 1 | Optimization of bisguanidinium-catalyzed asymmetric sulfoxidation of 2a.**

| Entry | [Mo] (5 mol%) | Additive (x equiv.) | Time (h) | Yield (%)* | ee (%)† |
|---|---|---|---|---|---|
| 1 | $(NH_4)_6Mo_7O_{24} \cdot 4H_2O$ | — | 24 | 15 | 0 |
| 2‡ | $(NH_4)_6Mo_7O_{24} \cdot 4H_2O$ | $CH_3CO_2H$ (1.0) | 24 | 0 | 0 |
| 3 | $(NH_4)_6Mo_7O_{24} \cdot 4H_2O$ | $CH_3CO_2H$ (1.0) | 24 | 14 | 20 |
| 4 | $(NH_4)_6Mo_7O_{24} \cdot 4H_2O$ | $CF_3CO_2H$ (1.0) | 8 | 80 | 4 |
| 5 | $(NH_4)_6Mo_7O_{24} \cdot 4H_2O$ | $NaHSO_4$ (1.0) | 19 | 99 | 69 |
| 6 | $(NH_4)_6Mo_7O_{24} \cdot 4H_2O$ | $KHSO_4$ (1.0) | 19 | 99 | 75 |
| 7 | $(NH_4)_6Mo_7O_{24} \cdot 4H_2O$ | $LiH_2PO_4$ (1.0) | 19 | 50 | 40 |
| 8 | $(NH_4)_6Mo_7O_{24} \cdot 4H_2O$ | $Na_2HPO_4$ (1.0) | 19 | 28 | 0 |
| 9 | $Li_2MoO_4$ | $KHSO_4$ (1.0) | 3 | 85 | 88 |
| 10 | $K_2MoO_4$ | $KHSO_4$ (1.0) | 2 | 99 | 86 |
| 11 | $Na_2MoO_4 \cdot 2H_2O$ | $KHSO_4$ (1.0) | 2 | 99 | 88 |
| 12 | $Na_2MoO_4 \cdot 2H_2O$ | $KHSO_4$ (0.5) | 2 | 99 | 89 |
| 13 | $Na_2MoO_4 \cdot 2H_2O$ | $KHSO_4$ (0.25) | 2 | 99 | 83 |
| 14§ | $Na_2MoO_4 \cdot 2H_2O$ | $KHSO_4$ (0.5) | 1 | 99 | 93 |
| 15§,‖ | $Na_2MoO_4 \cdot 2H_2O$ | $KHSO_4$ (0.5) | 1 | 99 | 94 |

Conditions: reaction was performed with 0.05 mmol of **2a**, $H_2O_2$ (1.05 equiv.), 5 mol% [Mo] source, 1 mol% (S,S)-**1a** in 1.0 ml of solvent.
*Yield of the isolated product.
†Determined by high-performance liquid chromatography analysis using CHIRALPAK AD-H column.
‡Without (S,S)-**1a**.
§$^iPr_2O$ as the solvent.
‖Reaction was performed with 0.2 mmol of **2a** at 0 °C with 2.5 mol% $Na_2MoO_4 \cdot 2H_2O$ with 99% isolated yield without sulfone byproduct.

temperature to 0 °C and using just 1 mol% of (S,S)-**1a** together with 2.5 mol% $Na_2MoO_4 \cdot 2H_2O$ in $^iPr_2O$, affording 2-sulfinyl acetate **3a** in 99% yield with 94% ee (entry 15). The absolute configuration of **3a** was confirmed to be S through comparison with the reported data[46].

**Substrate scope of various sulfides using (S,S)-1a.** The reaction scope was examined using a series of substrates with a relatively low reactivity, in which the electron density of sulfur is reduced, due to strong electron-withdrawing groups like ester, ketone and nitrile (Table 2). The reactions performed efficiently and were generally completed within 1 h. For benzyl 2-sulfanylacetates with different substituents on the aromatic ring, dialkyl sulfoxides **3b**–**3i** were obtained in high yields and excellent enantioselectivities. Sulfoxide **3j** bearing 2-thienyl was obtained in high yield and good enantioselectivity without oxidation at the thiophene. With a slight variation of reaction conditions, using 0.25 equivalent of $KHSO_4$ and 2.5 mol% of $K_2MoO_4$, various aromatic 2-sulfanylacetates were efficiently converted to alkyl aryl sulfoxides **3k**–**3q** in high yields with good enantioselectivities. For the oxidation of sulfide-bearing para-OMe substituent, leading to sulfoxide **3l**, slight over-oxidation to sulfone was observed. For a less reactive substrate **2r**, the reaction was conducted at room temperature using 1.5 equiv. $H_2O_2$, affording sulfoxide **3r** with good enantioselectivity. With less favourable substrates such as tert-butyl substituted 2-sulfanyl acetate **2s**, low enantioselective induction was observed (Table 2, **3s**).

To further explore the scope, a diverse range of substrates bearing different functional groups were examined (Table 2, **3t**–**3y**). 3-Sulfinyl propanoate **3t** was produced with excellent enantioselectivity. Sulfoxides **3u**–**3y** bearing amide, ketone, acrylate, nitrile and aldehyde moieties were furnished with good to excellent enantioselectivities. The absolute configurations of **3f** and **3o** were confirmed to be R and S, respectively, using single-crystal X-ray diffraction; thus, absolute configurations of sulfoxides **3** were assigned by analogy to either **3f** or **3o**. The practical utility was successfully demonstrated using a

gram-scale synthesis of (R)-modafinil (armodafinil), a commercial drug, using 0.25 mol% of (R,R)-**1a** (Fig. 3).

**Identification and characterization of ion pair (R,R)-1b.** We attempted to identify the reactive catalytic species by mimicking the reaction conditions in the absence of sulfide substrate (Fig. 4). After a simple workup procedure, (R,R)-**1b** was isolated and a single crystal suitable for X-ray diffraction was grown by vapour diffusion of $Et_2O$ into a dimethylformamide (DMF) solution of (R,R)-**1b**. The structure of (R,R)-**1b** was fully characterized using X-ray analysis (Fig. 2b), $^{95}Mo$ nuclear magnetic resonance (NMR) (Fig. 5b) and fourier transform-infrared spectroscopy (FT-IR) (Supplementary Fig. 2).

The achiral anionic metallic species $[(\mu-SO_4)Mo_2O_2(\mu-O_2)_2(O_2)_2]^{2-}$ is revealed by X-ray crystallography to be embedded within the chiral cavity formed by two side arms of the chiral bisguanidinium dication (Fig. 2b). The coordination geometry surrounding the Mo was clearly elucidated (Fig. 5a). The $SO_4^{2-}$ ligand plays a crucial role in constructing the dimeric symmetric structure. Each Mo centre comprises one bridging peroxo ligand, one side-on peroxo group and a terminal oxo ligand, with the sulfate group acting as a bipodal ligand to the two Mo atoms. Each Mo atom is 7-coordinated with oxygen atoms in a pentagonal bipyramidal arrangement. The two associated pentagonal bipyramids share one edge $[O_9–O_{10}]$ and the two Mo atoms are connected by two $\mu–\eta^1:\eta^2$ peroxo-bridges, $[O_8–O_9$ and $O_{11}–O_{10}]$. Both $Mo_1–O_5$ and $Mo_2–O_{12}$ bonds have the same length (1.659(7) Å) that falls in a typical range for the Mo = O bond. Generally, the bridging peroxo $O_8–O_9$ (1.482(9) Å) and $O_{10}–O_{11}$ (1.473(10) Å) bond lengths are slightly longer than the other side-on peroxo $O_6–O_7$ (1.458(10) Å) and $O_{13}–O_{14}$ (1.467(10) Å) bond lengths. $^{95}Mo$ NMR spectrum of (R,R)-**1b** was also obtained in DMF-$d_7$ at 22 °C, using 2 M $Na_2MoO_4 \cdot 2H_2O$ solution in $D_2O$ as an external reference (assigned to 0 p.p.m.). The chemical shift at $-199.3$ p.p.m. is characteristic of oxodiperoxomolybdate species (Fig. 5b)[52].

We found that (R,R)-**1b** (1.0 equiv.), prepared using the method in Fig. 4, can be used directly as the oxidant for

## Table 2 | Substrate scope of sulfides in asymmetric sulfoxidation.

$$R_1\text{-S-}R_2 \xrightarrow[\substack{\text{KHSO}_4\ (0.5\ \text{equiv.}), \\ {}^i\text{Pr}_2\text{O}\ (4\ \text{mL}),\ 0^\circ\text{C},\ 1\ \text{h}}]{\substack{(S,S)\text{-}1a\ (1\ \text{mol}\%) \\ \text{Na}_2\text{MoO}_4\cdot 2\text{H}_2\text{O}\ (2.5\ \text{mol}\%) \\ 35\%\ \text{aq.\ H}_2\text{O}_2\ (1.05\ \text{equiv.})}} R_1\text{-S(O)-}R_2$$

**2** 0.2 mmol → **3**

**3b**, 99% yield, 90% ee

**3c**, 92% yield, 92% ee

**3d**, 96% yield, 83% ee*

**3e**, 94% yield, 96% ee

**3f**, 92% yield, 91% ee

**3g**, 98% yield, 92% ee

**3h**, 98% yield, 93% ee

**3i**, 99% yield, 93% ee

**3j**, 94% yield, 89% ee*

**3k**, 91% yield, 86% ee[†,‡]

**3l**, 94% yield, 79% ee[†,‡]

**3m**, 93% yield, 89% ee*[,†,‡]

**3n**, 93% yield, 90% ee[†,‡]

**3o**, 95% yield, 91% ee[†,‡]

**3p**, 99% yield, 89% ee*[,†,‡]

**3q**, 97% yield, 83% ee[†,‡]

**3r**, 79% yield, 74% ee[§]

**3s**, 83% yield, 37% ee*

**3t**, 88% yield, 94% ee[†]

**3u**, 96% yield, 82% ee[†]

**3v**, 99% yield, 90% ee

**3w**, 84% yield, 77% ee[†]

**3x**, 87% yield, 80% ee

**3y**, 82% yield, 65% ee[†,‡]

Conditions: reaction was performed with 0.2 mmol of **2**, $H_2O_2$ (1.05 equiv.), KHSO$_4$ (0.5 equiv.), 2.5 mol% Na$_2$MoO$_4$·2H$_2$O, 1 mol% (S,S)-**1a** in 4.0 ml of solvent at 0 °C.
*$^n$Bu$_2$O as the solvent.
†K$_2$MoO$_4$ as [MoO$_4$]$^{2-}$ source.
‡0.25 equiv. of KHSO$_4$.
§The reaction was conducted at room temperature for 24 h using 1.5 equiv. of 35% aqueous H$_2$O$_2$.

**Figure 3 | Gram-scale synthesis of R-modafinil (armodafinil).** Reaction conditions: (i) (R,R)-**1a** (0.25 mol%), Na$_2$MoO$_4$·2H$_2$O (2.5 mol%), 35% aq. H$_2$O$_2$ (1.05 equiv.), KHSO$_4$ (0.5 equiv.), $^n$Bu$_2$O (0.05 M), rt, 8 h; (ii) NH$_3$ (10.0 equiv., 2 M in MeOH), rt, 24 h.

sulfoxidation, without additional aqueous H$_2$O$_2$, providing sulfoxide ***ent*-3a** in 90% yield and 80% *ee* in 0.5 h (Fig. 6a, equation 1). This result indicates that (R,R)-**1b** is the actual oxidizing specie providing high enantiodiscrimination. Utilizing

0.25 equiv. (R,R)-**1b** led to the formation of ***ent*-3a** in 50% yield in 24 h with 31% *ee* (Fig. 6a, equation 2), demonstrating that two out of four peroxo moieties on (R,R)-**1b** are active oxygen donors, as two equivalent of active oxygen from (R,R)-**1b** are transferred to

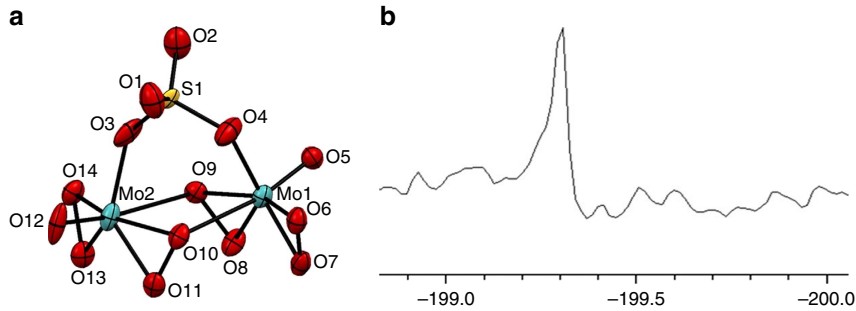

**Figure 4 | Preparation of [BG]$^{2+}$[(μ-SO$_4$)Mo$_2$O$_2$(μ-O$_2$)$_2$(O$_2$)$_2$]$^{2-}$ (R,R)-1b.** Conditions: Na$_2$MoO$_4$·2H$_2$O (2.5 mol%), 35% aq. H$_2$O$_2$ (1.0 equiv.), KHSO$_4$ (0.5 equiv.) or H$_2$SO$_4$ (0.25 equiv.), Et$_2$O (2 ml), rt, 2 h.

**Figure 5 | Characterization of the anionic cluster [(μ-SO$_4$)Mo$_2$O$_2$(μ-O$_2$)$_2$(O$_2$)$_2$] in (R,R)-1b.** (a) ORTEP view of [(μ-SO$_4$)Mo$_2$O$_2$(μ-O$_2$)$_2$(O$_2$)$_2$]$^{2-}$ dianion in (R,R)-1b with the atom numbering scheme. (b) $^{95}$Mo NMR spectrum of (R,R)-1b in DMF-d$_7$ (0.05 M, 22 °C).

**Figure 6 | Mechanistic insights.** (a) [BG]$^{2+}$[(μ-SO$_4$)Mo$_2$O$_2$(μ-O$_2$)$_2$(O$_2$)$_2$]$^{2-}$ (R,R)-1b as the sole oxidant. (b) 1 mol% (R,R)-1b as catalyst.

the sulfides[53]. The second oxygen transfer is slower and less enantioselective than the first. As expected, (R,R)-1b can be used catalytically in the presence of H$_2$O$_2$, providing sulfoxide **ent**-3a in 95% yield with 91% ee at a loading of 1 mol% (Fig. 6b, equation 3). This result is comparable to the reaction, in which this active catalyst is prepared in situ from (R,R)-1a (Fig. 3). The catalyst can be recycled from the reaction (Fig. 6b, equation 3) and used for a second round of reaction, but an additional amount of 0.5 equiv. of KHSO$_4$ must be added to restore the reactivity and enantioselectivity (Fig. 6b, equation 4).

Computational studies of (R,R)-1b using the ONIOM method revealed that a more stable ion-pairing arrangement, with the distance between the anionic cluster and the cationic bisguanidinium noticeably reduced compared with that in the crystal structure (Fig. 7). As a result of this rearrangement, approach of the substrate to most of the peroxo-oxygen atoms is obstructed by bisguanidinium and sulphate groups. Only one of side-on peroxo-oxygen atoms (marked as O$_{14}$ in Fig. 7) remains accessible for reaction. This is consistent with the experimentally observed high enantioselectivity, since restricted access to secondary reaction sites will result in a reaction with higher selectivity.

Reaction selectivity was evaluated using 50 mol% tetrabutylammonium hydrogen sulfate ($^n$Bu$_4$NHSO$_4$) as an achiral ion-pairing reagent (Fig. 8). A high level of enantiocontrol can still be achieved, which indicates ion pairing interaction between chiral bisguanidinium and dinuclear oxodiperoxomolybdosulfate anion [(μ-SO$_4$)Mo$_2$O$_2$(μ-O$_2$)$_2$(O$_2$)$_2$]$^{2-}$ accelerates the reaction rate significantly over tetrabutylammonium, promoting the

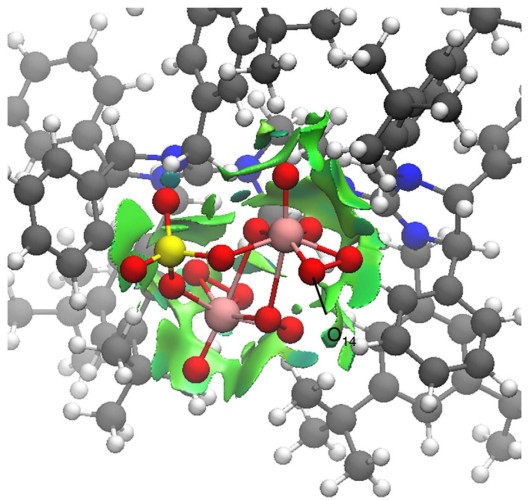

**Figure 7 | Geometry optimization of ion pair (R,R)-1b with ONIOM method.** Structure of (R,R)-1b obtained from ONIOM geometry optimization, where atoms are color-coded as follows: C (grey), N (blue), H (white), S (yellow), O (red) and Mo (pink). The displayed NCI surface of bisguanidinium indicates interactions between the anionic cluster [(μ-SO$_4$)Mo$_2$O$_2$(μ-O$_2$)$_2$(O$_2$)$_2$] and bisguanidinium.

reaction through the desired asymmetric pathway. The stereo-induction observed in the current methodology may be ascribed to ion pairing interaction and other non-covalent interactions

**Figure 8 | Effect of achiral tetrabutylammonium cation.** Evaluation of selectivity using 50 mol% of $^nBu_4NHSO_4$ as achiral ion-pairing reagent with 1 mol% of chiral catalyst (R,R)-**1a**.

between dicationic bisguanidinium, oxodiperoxomolybdosulfate anion, as well as the substrates in the stereoselectivity determining transition state (Supplementary Fig. 5)[33,54].

## Discussion

In the present study, we have described the first catalytic use of peroxomolybdate for enantioselective sulfoxidation; a series of enantioenriched dialkyl sulfoxides and alkyl aryl sulfoxides have been obtained using inexpensive molybdates and aqueous $H_2O_2$ through a simple experimental protocol. The practical value of current methodology was demonstrated using a gram-scale synthesis of armodafinil, a commercial drug, with a low loading (0.25 mol%) of bisguanidinium. The 'active' catalyst is isolatable, stable and has been identified to be bisguanidinium dinuclear oxodiperoxomolybdosulfate ion pair $[\mathbf{BG}]^{2+}[(\mu\text{-}SO_4)Mo_2O_2(\mu\text{-}O_2)_2(O_2)_2]^{2-}$. Its structure is also unambiguously confirmed by X-ray analysis.

## Methods

**General information.** The synthesis of sulfide substrates **2** are provided in Supplementary Note 1. For the details of mechanistic studies, see Supplementary Figs 3–5, Supplementary Tables 1–2 and Supplementary Note 2. For details of X-ray analysis, see Supplementary Figs 6–9, Supplementary Methods and Supplementary Data 2–9. For details of computational studies, see Supplementary Figs 10–13, Supplementary Tables 3–5, Supplementary Methods and Supplementary Data 1. For the $^1H$, $^{13}C$, $^{19}F$ and $^{95}Mo$ NMR data, and spectra of the compounds in the article, see Supplementary Figs 14–70 and Supplementary Methods. For the high-performance liquid chromatography spectra of sulfoxide product **3** and **4** in this article, see Supplementary Figs 71–96.

**Preparation of ion pair (R,R)-1b.** To the solution of $Na_2MoO_4 \cdot 2H_2O$ (24.1 mg, 0.1 mmol) dissolved in 1 M $H_2SO_4$ (1 ml), 35% $H_2O_2$ (345 µl, 4.0 mmol) was added dropwise to give a yellow solution at room temperature. Then the above solution was added dropwise to a solution of (R,R)-**1a** (56.4 mg, 0.04 mmol) in $Et_2O$ (2 ml). After vigorously stirring for 15 min, a pale yellow precipitate was formed in the $Et_2O$ layer. After further stirring for 2 h and removal of $Et_2O$ by evaporation, 4 ml deionized water was added and the resulting heterogeneous mixture was submitted to ultrasound for 1 min. Then the pale yellow solid was filtered off and washed with deionized water (40 ml). After drying over concentrated $H_2SO_4$ under vacuum at room temperature, (R,R)-**1b** was obtained as a pale yellow powder (65.5 mg, 91% yield) and its structure was characterized and determined by X-ray single-crystal diffraction. Increase of the amount of $Na_2MoO_4 \cdot 2H_2O$ to 0.1 equivalent or replacement of 1 M $H_2SO_4$ with 0.5 equivalent of solid $KHSO_4$ all led to the formation of identical complex (R,R)-**1b**, which is confirmed by X-ray diffraction analysis.

**General procedure for synthesis of sulfoxides 3.** A 10 ml round-bottom flask (RBF) was charged with a solution of sulfide **2** (0.2 mmol) and bisguanidinium phase-transfer catalyst (S,S)-**1a** (2.8 mg, 0.002 mmol) in $^iPr_2O$ (4 ml). Then $Na_2MoO_4 \cdot 2H_2O$ (1.2 mg, 0.005 mmol) and $KHSO_4$ (13.6 mg, 0.1 mmol) were added. The reaction mixture was stirred for 5 min in an ice bath, and then aqueous 35% $H_2O_2$ (18.1 µl, 0.21 mmol) was added in one portion. The resulting mixture was stirred vigorously at 0 °C and monitored by thin-layer chromatography until **2** was completely consumed. Purification using silica gel column chromatography ($CH_2Cl_2/EtOAc = 2:1$) afforded the desired sulfoxide **3**. Minor changes in the amount of $KHSO_4$ and choice of molybdate salt ($K_2MoO_4$) and solvent ($^nBu_2O$) were conducted for some substrates to achieve slightly better enantioselectivity.

**Gram-scale experiment for synthesis of armodafinil 4.** A 250 ml round-bottom flask was charged with a solution of methyl 2-(benzhydrylthio)acetate **2a** (1.36 g, 5 mmol) and bisguanidinium phase-transfer catalyst (R,R)-**1a** (17.6 mg, 0.0125 mmol) in $^nBu_2O$ (100 ml). Then $Na_2MoO_4 \cdot 2H_2O$ (30 mg, 0.125 mmol), $KHSO_4$ (340 mg, 2.5 mmol) and 35% aq. $H_2O_2$ (453 µl, 5.25 mmol) were added at

room temperature. The resulting mixture was stirred vigorously and monitored by thin-layer chromatography and **2a** was completely consumed within 8 h. Purification using silica gel column chromatography ($CH_2Cl_2/EtOAc = 2:1$) afforded the sulfoxide product ent-**3a** with R configuration, 1.32 g, 91% yield, 91% ee. Then the obtained sulfoxide (1.32 g, 4.58 mmol) was treated with 2 M ammonical methanol (23 ml, 10.0 equiv.) and the resulting solution was stirred at room temperature for 24 h. Purification using silica gel column chromatography ($CH_2Cl_2/MeOH = 20:1$) afforded **4** as a white solid, 1.19 g, 95% yield, 91% ee.

**Data availability.** CCDC 1456987–1456990 contain the supplementary crystallographic data for this paper. These data can be obtained free of charge from The Cambridge Crystallographic Data Centre via www.ccdc.cam.ac.uk/data_request/cif. The data that support the findings of this study are available from the corresponding authors on request.

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

## Acknowledgements

We acknowledge Nanyang Technological University (M4080946.110, M4011372.110) for financial support. We thank Ms E.L. Goh for her technical support in ⁹⁵Mo NMR analysis.

## Author contributions

L.Z. and C.-H.T designed and conceived the project. L.Z. performed the experiments and analysed the experimental results. C.-H.T and L.Z wrote the manuscript. C.W. and X.Y. contributed to the initial studies. A.M.P.M. and H.H. performed the computational studies. R.G. and Y.L. contributed to the X-ray data collection.

## Additional information

**Competing financial interests:** The authors declare no competing financial interests.

