## [Peer Review file · Nature Communications]

Reviewer #1 (Remarks to the Author):

In this manuscript, Tan and coworkers report the enantioselective sulfoxidation of 2-sulfanyl acetate in the presence of chiral ion pair generated in situ from bisguanidinium and Na₂MoO₄·2H₂O using H₂O₂ as terminal oxidant. The method is operationally simple, and the products are obtained in high yields with excellent enantioselectivities. Further, structural characterization of the ion pair catalyst is achieved using single crystal X-ray crystallography, which reveals the role of KHSO₄. Given the novelty of the work and also the advance reported here, this reviewer strongly supports the publication of this manuscript in Nature Commun.

Minor points:

1. The compound number 3g in page 5 should be corrected as 3j (line 6, paragraph 2).
2. The sentence "For less reactive substrate 2r, ...3r with good enantioselectivities." (line 2, paragraph 1, page 6) should be corrected as "For less reactive substrate 2r, ...3r with good enantioselectivity."
3. The author described "...low enantioselective induction was observed (Table 2, entry 10)". However, I did not find where "Table 2" is, it may be "...low enantioselective induction was observed (Scheme 1, 3s)."
4. In this manuscript, the authors demonstrate the practical utility for a gram-scale synthesis of (R)-modafinil (armodafinil). But the ee value is 91%. Can the ee value be improved by recrystallization?

Reviewer #2 (Remarks to the Author):

This is an asymmetric preparation of sulfoxide using Mo catalyst. The substrate needs carbonyl group at beta position for higher enantioselectivity. This is relatively similar to the Ti catalyzed reaction of hydroxy sulfoxide synthesis (JACS 137, 15612-15615 (2015)). Thus, both method seems to be the similar concept of substrate controlled reaction. The reaction seems to proceed smoothly under the conditions described. However, I am wondering the second oxidation from the product of sulfoxide to sulfone does not proceed at all or not. If the sulfone formation proceeds, the kinetic resolution should take place which significantly influence the final results of %ee of sulfoxide. Overall, I think this paper is relatively good asymmetric reaction and may be useful in future.

Reviewer #3 (Remarks to the Author):

“Bisguanidinium Dinuclear Oxodiperoxomolybdatesulfate Ion Pair-Catalyzed Enantioselective Sulfoxidation”

The Authors describe a new Ion Pair catalyst for enantioselective sulfoxidation. They optimize the catalytic conditions and investigate the scope of the system on a host of substrates. The crystallography in this paper is used for characterization and to help elucidate the nature of the active catalyst. The structural details of the molybdenum cluster are well described in the paper.

Three of the structures were organic and included in the supporting information, though I see no mention of the data in the main manuscript. There are several errors and omissions in the presentation of the X-ray data, that must be addressed prior to publication.

There is only a very short sentence in the experimental section describing the instrument used for data collection. It does not indicate which software programs were used, which absorption correction was used or even which X-ray source is used. This ambiguity in X-ray source also arises in the structural data (S31 – 34) as the extinction coefficient is reported as $\mu(\text{CuK}\alpha)$ and the wavelength is reported as 0.71074 Å, which is the wavelength for Mo radiation. A more robust explanation of the X-ray experiment is required. The Authors should make sure to double-check which source was used and subsequently revise the manuscript

Several structures, including the main catalyst of interest, contained disorder. Handling of disorder should be described either embedded in the CIF file or, ideally, in the supporting information. This information is presently absent and should be added. It appears, from reading the .res files, that most of the disorder was modeled correctly, but a short paragraph for each disordered molecule on which restraints were used and why is important.

It appears in the structure CCDC 1456989 (S)-3o that a RIGU restraint was applied globally rather than to a particular disorder. Applying such a global restraint seems excessive, especially without explanation. This structure also appears to be twinned and the symmetry is close to that of $c/2$. Double check that you have the correct space group and/or add explanation as to your handling of the data.

Adding the Flack parameter and e.s.d. to the supporting paragraph for each molecule will be helpful for readers.

I can recommend the crystallographic portion of this manuscript for publication in Nature Communications after these explanations and clarifications are fully addressed.

Dear reviewers,

We thank you for the opportunity to improve our manuscript. These are the corrections that we made.

Reviewer #1 (Remarks to the Author):

In this manuscript, Tan and coworkers report the enantioselective sulfoxidation of 2-sulfanyl acetate in the presence of chiral ion pair generated in situ from bisguanidinium and $\text{Na}_2\text{MoO}_4 \cdot 2\text{H}_2\text{O}$ using H_2O_2 as terminal oxidant. The method is operationally simple, and the products are obtained in high yields with excellent enantioselectivities. Further, structural characterization of the ion pair catalyst is achieved using single crystal X-ray crystallography, which reveals the role of KHSO_4 . Given the novelty of the work and also the advance reported here, this reviewer strongly supports the publication of this manuscript in Nature Commun.

Minor points:

1. The compound number 3g in page 5 should be corrected as 3j (line 6, paragraph 2).

Reply: We have changed **3g** to **3j** (page 5).

2. The sentence "For less reactive substrate 2r, ...3r with good enantioselectivities." (line 2, paragraph 1, page 6) should be corrected as "For less reactive substrate 2r, ...3r with good enantioselectivity."

Reply: We have made the correction as suggested (page 6).

3. The author described "...low enantioselective induction was observed (Table 2, entry 10)". However, I did not find where "Table 2" is, it may be "...low enantioselective induction was observed (Scheme 1, 3s)".

Reply: We have corrected the sentence to "...low enantioselective induction was observed (Scheme 1, **3s**)."

4. In this manuscript, the authors demonstrate the practical utility for a gram-scale synthesis of (R)-modafinil (armodafinil). But the ee value is 91%. Can the ee value be improved by recrystallization?

Reply: The ee values of (R)-modafinil can to be improved by recrystallization in absolute ethanol. The protocol is reported in the patent (Rebiere, F., DURET, G. & Prat, L. Process for enantioselective synthesis of single enantiomers of modafinil by asymmetric oxidation. WO2005028428A1 (2005)), page 45, table 21, example 18). Initial ee values of 91.6% can be improved to 99.1% after single recrystallization.

Reviewer #2 (Remarks to the Author):

This is an asymmetric preparation of sulfoxide using Mo catalyst. The substrate needs carbonyl group at beta position for higher enantioselectivity. This is relatively similar to the Ti catalyzed reaction of hydroxy sulfoxide synthesis (JACS 137, 15612-15615

(2015)). Thus, both method seems to be the similar concept of substrate controlled reaction. The reaction seems to proceed smoothly under the conditions described.

However, I am wondering the second oxidation from the product of sulfoxide to sulphone does not proceed at all or not. If the sulfone formation proceeds, the kinetic resolution should take place which significantly influence the final results of %ee of sulfoxide. Overall, I think this paper is relatively good asymmetric reaction and may be useful in future.

Reply: Our observations indicated that in general, the sulfide was completely consumed within 1 hour in the presence of 1.05 equivalent of hydrogen peroxide. The sulfoxide was isolated with high yield (79-99%). The second oxidation from sulfoxide to sulphone did not proceed except for substrate **2l**. We have previously indicated this in the manuscript (page 5, paragraph 2, lines 11-12).

Reviewer #3 (Remarks to the Author):

The Authors describe a new Ion Pair catalyst for enantioselective sulfoxidation. They optimize the catalytic conditions and investigate the scope of the system on a host of substrates. The crystallography in this paper is used for characterization and to help elucidate the nature of the active catalyst. The structural details of the molybdenum cluster are well described in the paper.

Three of the structures were organic and included in the supporting information, though I see no mention of the data in the main manuscript. There are several errors and omissions in the presentation of the X-ray data, that must be addressed prior to publication.

Reply: We have added extra sentences in the revised manuscript (pages 6 and 7) to guide the readers to the three organic structures - “The absolute configurations of **3f** and **3o** were confirmed to be *R* and *S* respectively, using single-crystal X-ray diffraction; thus, the absolute configurations of sulfoxides **3** were assigned by analogy to either **3f** or **3o** (see Supporting Information)”.

There is only a very short sentence in the experimental section describing the instrument used for data collection. It does not indicate which software programs were used, which absorption correction was used or even which X-ray source is used.

Reply: The detail for instrument, software programs, absorption correction and X-ray source used for data collection and structural refinement have been presented in the revised SI, page S31. The details are as follows: For instrument, intensity data for compounds (*R,R*)-**1b**, (*R*)-**3f**, (*S*)-**3o** and (*R*)-**4** were collected using a Bruker Kappa APEX II diffractometer. For software programs, the structure was solved by SHELXT (SHELXL-2014/7 (Sheldrick, 2014)) and refined for all data by full-matrix least-squares methods on F^2 . For absorption correction, all data were corrected for absorption effects using the Multi-Scan method (SADABS). For X-ray source, all the four compounds (*R,R*)-**1b**, (*R*)-**3f**, (*S*)-**3o** and (*R*)-**4** are analyzed with the Mo X-ray source (0.71073 Å).

This ambiguity in X-ray source also arises in the structural data (S31 – 34) as the extinction coefficient is reported as $\mu(\text{CuK}\alpha)$ and the wavelength is reported as 0.71074 Å, which is the wavelength for Mo radiation. A more robust explanation of the X-ray experiment is required. The Authors should make sure to double-check which source was used and subsequently revise the manuscript.

Reply: We have carefully checked the X-ray experiment; Mo X-ray source (0.71073 Å) is used for the analysis of all compounds. Correction has been made in the revised supporting information (S31-S35).

Several structures, including the main catalyst of interest, contained disorder. Handling of disorder should be described either embedded in the CIF file or, ideally, in the supporting information. This information is presently absent and should be added. It appears, from reading the .res files, that most of the disorder was modeled correctly, but a short paragraph for each disordered molecule on which restraints were used and why is important.

Reply: We have added the information of handling of disorder for two disordered molecules (*R,R*-**1b** and *R*-**3f** in the revised supporting information (S31-S33). It is indicated as follows: For compound (*R,R*-**1b**, four *t*-Butyl groups and one DMF molecule were disordered over two positions. The four *t*-Butyl groups (C5-C8, C11-C14, C40-C43, and C90-C93) were modelled with restraints (SAME, RIGU C39 > C43a C10 > C14a C4 > C8a C89 > C93a; SIMU 0.01 C40 > C43a C11 > C14a C5 > C8a C90 > C93a; ISOR 0.01 C5a > C8a) to restrain the groups having similar geometry and anisotropic displacement parameters. The DMF molecule (C102 to O17) was modeled with restraints (SAME; FLAT C102 > O17; FLAT C2a > O17a; RIGU C102 > O17a; SIMU 0.01 C102 > O17a) to restrain the two components having similar geometry and anisotropic displacement parameters. For compound (*R*-**3f**, the ester group C8 to C13, O2 and O3 was disordered over two positions and modelled with restraints (SAME, RIGU O2 > C13a SIMU 0.02 O2 > C13a and SADI C8 S1 C8a S1) to restrain the two components having similar geometry and anisotropic displacement parameters. For compounds (*S*-**3o** and *R*-**4**, no disorder was included.

It appears in the structure CCDC 1456989 (*S*-**3o**) that a RIGU restraint was applied globally rather than to a particular disorder. Applying such a global restraint seems excessive, especially without explanation. This structure also appears to be twinned and the symmetry is close to that of *c*/2. Double check that you have the correct space group and/or add explanation as to your handling of the data.

Reply: We have checked the space group and added the explanation of applying a RIGU restraint globally in revised supporting information (S34). It is indicated as follows: The crystal for (*S*-**3o**) is twinned with the twin law being 1 0 1 0 -1 0 0 0 -1 and the BASF was refined to 0.294. The restraint RIGU was applied globally to prevent two carbon atoms C20 and C22 to be NPD; if applied to the group containing C20 and C22 atoms, C22 would show very high ADP max/min ratio. The structure

could not be solved in space group $C2$ or $C2/m$ and the Flack parameter of 0.008(8) in $P2_1$ could be an indication that the space group is correct.

Adding the Flack parameter and e.s.d. to the supporting paragraph for each molecule will be helpful for readers. I can recommend the crystallographic portion of this manuscript for publication in Nature Communications after these explanations and clarifications are fully addressed.

Reply: We have added the Flack parameters and e.s.d. of all the four molecules in the revised supporting information S32-S35. Flack parameters and e.s.d. of all the four compound are shown as below and the e.s.d values are in the brackets.

(R,R)-**1b**: 0.00(2)

(R)-**3f**: 0.04(4)

(S)-**3o**: 0.008(8)

(R)-**4**: 0.06(3)

Yours sincerely,
Choon-Hong Tan